# Therapeutic Plasma Exchange in Certain Immune-Mediated Neurological Disorders: Focus on a Novel Nanomembrane-Based Technology

**DOI:** 10.3390/biomedicines11020328

**Published:** 2023-01-25

**Authors:** Dimitar G. Tonev, Albena B. Momchilova

**Affiliations:** 1Department of Anesthesiology and Intensive Care, Medical University of Sofia, University Hospital “Tzaritza Yoanna—ISUL”, 1527 Sofia, Bulgaria; 2Institute of Biophysics and Biomedical Engineering, Bulgarian Academy of Science, 1113 Sofia, Bulgaria

**Keywords:** nanomembrane, therapeutic plasma exchange, immune-mediated neurological disorders, efficacy, cost-effectiveness, oxidative stress, neuroregeneration

## Abstract

Therapeutic plasma exchange (TPE) is an efficient extracorporeal blood purification technique to remove circulating autoantibodies and other pathogenic substances. Its mechanism of action in immune-mediated neurological disorders includes immediate intravascular reduction of autoantibody concentration, pulsed induction of antibody redistribution, and subsequent immunomodulatory changes. Conventional TPE with 1 to 1.5 total plasma volume (TPV) exchange is a well-established treatment in Guillain-Barre Syndrome, Chronic Inflammatory Demyelinating Polyradiculoneuropathy, Neuromyelitis Optica Spectrum Disorder, Myasthenia Gravis and Multiple Sclerosis. There is insufficient evidence for the efficacy of so-called low volume plasma exchange (LVPE) (<1 TPV exchange) implemented either by the conventional or by a novel nanomembrane-based TPE in these neurological conditions, including their impact on conductivity and neuroregenerative recovery. In this narrative review, we focus on the role of nanomembrane-based technology as an alternative LVPE treatment option in these neurological conditions. Nanomembrane-based technology is a promising type of TPE, which seems to share the basic advantages of the conventional one, but probably with fewer adverse effects. It could play a valuable role in patient management by ameliorating neurological symptoms, improving disability, and reducing oxidative stress in a cost-effective way. Further research is needed to identify which patients benefit most from this novel TPE technology.

## 1. Introduction

Therapeutic plasma exchange (TPE) has become a well-established therapeutic procedure in neurological clinical practice for numerous pathological conditions. Guillain-Barre Syndrome (GBS), Chronic Inflammatory Demyelinating Polyradiculoneuropathy (CIDP), Neuromyelitis Optica Spectrum Disorder (NMOSD), Myasthenia Gravis (MG) and Multiple Sclerosis (MS) are among the five most frequent neurological indications for TPE [1,2]. Moreover, they are among the most common immune-mediated disorders with indications for urgent TPE in the intensive care setting as well [3]. Currently, TPE is recommended 5–7 times within 10–14 days, with 1 to 1.5 total plasma volume (TPV) exchange per procedure per day, including these five disorders [4]. A few studies and one review article indicate similarly, that the effectiveness of TPE may be also given with volumes below this recommendation (i.e., less than 1 TPV exchange), the so-called low volume plasma exchange (LVPE). The LVPE was achieved by means of conventional TPE, immuno-absorption technique, or by using a novel nanomembrane-based TPE technology [5,6,7,8,9]. There is insufficient evidence for the efficacy of LVPE procedures implemented in the above neurological conditions, including their impact on conductivity and neuroregenerative recovery. Moreover, despite the growing scientific interest in the application of LVPE in the treatment of neurological diseases and in the field of clinical and experimental hepatology [10,11,12,13,14,15,16,17,18,19,20,21], only one study using a nanomembrane-based TPE technology was mentioned in the context of the LVPE approach feasibility [8,9]. The aim of this narrative review is to give more insights into the role of nanomembrane-based technology as an alternative LVPE treatment option in GBS, CIDP, NMOSD, MG, and MS, considering the modern TPE technologies in the context of current clinical practice and with reference to the recently published guidelines from the American Academy of Neurology (AAN) and the American Society for Apheresis (ASFA), where this debated topic is likely to be covered in the upcoming issues.

In addition, the last review on blood apheresis technologies was focused on the role of nanoparticles in improving membrane-based apheresis without commenting on other innovative nanotechnologies, such as our nanomembrane-based one (Table 1) [22]. It is obvious that a new focused review is necessary to fill the gap concerning this novel technology as well.

## 2. Methodology

A literature search was conducted through December 2022 of MEDLINE, EMBASE, and Cochrane Library, based on Medical Subject Heading (MeSH) “therapeutic plasma exchange”, “nanomembrane-based”, “plasmapheresis”, “apheresis”, immuno-mediated”, “autoimmune”, “neurological”, “disorders”, “diseases”, “efficacy”, “effectiveness”, “cost-effectiveness”, “cost-minimization”, “analysis”, “immunomodulation”, “oxidative stress”, “conductivity”, “regeneration”, “Guillain-Barre Syndrome”, “GBS”, “Chronic Inflammatory Demyelinating Polyradiculoneuropathy”, “CIDP”, “Neuromyelitis Optica Spectrum Disorder”, “NMOSD”, “Myasthenia Gravis”, “MG”, “Multiple Sclerosis”, “MS”, as well as by manual search in the local database. The search had no language restrictions. 

## 3. Principles of TPE 

The terms plasmapheresis and plasma exchange are often used interchangeably, but when correctly used, they mean different procedures. Plasmapheresis refers to procedures in which plasma is separated from the blood either by centrifugation or membrane filtration. It usually refers to the removal of a small volume of plasma, not more than 15% of TPV, without the need for volume replacement. Plasma exchange is a therapeutic intervention that involves extracorporeal removal, return, or exchange of blood plasma or components. The term means the removal of large volume of plasma (1 to 1.5 of patients TPV per treatment) with appropriate volume replacement using colloid solutions (e.g., albumin and/or fresh frozen plasma (FFP)) or a combination of crystalloid/colloid solutions [34]. TPE serves to remove pathogenic substance (large molecules such as autoantibodies, immune complexes, cytokines, etc.) from the intravascular space, which ensure its rapid onset of action. However, the mechanism of action of TPE in immune-mediated neurological disorders involves more than the removal of large molecules. For example, the use of TPE may also modulate cellular immunity by altering the ratio of T-helper type-1 (Th-1) and type-2 (Th-2) cells in peripheral blood in patients with Miller Fisher syndrome, a variant of acute inflammatory demyelinating polyradiculoneuropathy (AIDP, Guillian-Barre syndrome) [35]. Th-2 cells aid the humoral immune response by facilitating B-cell autoantibody production and play an important role in neurological autoimmune disorders. By shifting the peripheral T cells from a Th-2 predominance to a Th-1 predominance TPE modulates the pathogenic immune response and may exert a therapeutic effect within and beyond the time of TPE acute treatment. [36]. In the same immunomodulatory way, TPE increases T suppressor cell function in CIDP, which correlated with improvement in the patients’ neurological symptoms as measured on the Neurological Disability Score [37].

Centrifugation TPE involves the separation of blood components based on the specific gravity of the analyzed components. The obtained blood is placed in a centrifuge chamber, where the red cells which have higher density are first sedimented, followed by leucocytes and platelets [38]. Currently licensed TPE devices can operate with a continuous or an intermittent flow [39]. During the exchange, the patient’s separated plasma is discarded and replaced with a replacement fluid. 

Filtration TPE separates blood components based on the size of the analyzed particles. The plasma flows first through the filter, followed by platelets, erythrocytes, lymphocytes, and, finally, granulocytes. Most commonly the filtration involves passage through a hollow-fibre system in which the blood samples are pumped. During the exchange procedure, the plasma leaves the fibres through the side pores, and is consequently replaced by substitution fluid (Figure 1) [40,41]. 

An advanced variant of membrane plasmapheresis is the so-called cascade or double filtration plasmapheresis (DFPP) which by passing the whole blood through the first filter extracts the plasma. Then by passing the separated plasma through a second filter (with a side pore size 10-fold smaller than the first filter in order to retain only larger molecules such as immunoglobulin G (IgG) class proteins) the cascade mode eliminated IgG via the discarded plasma. The purified plasma is then returned to the patient minimizing the need for replacement fluids (Figure 2) [40,42].

TPE using centrifugal machines can be performed through peripheral or central venous access which could decrease the complications associated with the central venous access (infection, clotting with occlusion of the central venous line, pneumothorax, etc.) (Table 2). On the other hand, filtration machines are smaller, used lower extracorporeal volume, and are easier to use even for pediatric patients. The filtration technique is quicker and could exchange more plasma volume in a shorter period of time than the centrifugation one. However, heparin anticoagulation increases the risk of bleeding as a result of heparin-induced thrombocytopenia. In addition, the blood-filter interactions may activate the kallikrein-kinin system, creating an increase serum concentration of bradykinin with untoward hemodynamic effects, especially in patients using angiotensin-converting-enzyme inhibitors in the last 24 h before the TPE (a relative contraindication) [38]. 

Another advanced method of membrane plasmapheresis is the so-called immune-adsorption (IA) which by passing the whole blood through the first filter extracts the plasma. Then by passing the separated plasma through a second medical device (an adsorption column), the column active component (e.g., Staphylococcal protein A) specifically binds pathologic IgG and immune complexes and removes them up to 66%. The purified plasma is then returned to the patient avoiding the need of replacement fluids (Figure 3) [34,40].

In Table 2 are given some technical aspects of centrifugal separation and membrane filtration TPE techniques of relevance to neurological immune-mediated disorders. 

In the past, citrate was generally used for centrifugal-based TPE and heparin for membrane-based TPE [43,44,45]. According to the World Apheresis Registry, in which two-thirds of aphereses were therapeutic, 73% of procedures were provided with citrate anticoagulation [46,47]. 

## 4. Nanomembrane-Based TPE

The nanomembrane-based TPE is a non-selective method of blood purification that removes toxic and inflammatory blood components. It consists of passing the patient’s blood through several nanomembranes to filter large molecules of certain molecular weight [29,30,31,32,48,49,50]. The nanomembrane-based technology involves the use of the “Hemophenix” apparatus (Figure 4) with the ROSA nanomembrane (“Trackpore Technology”, Moscow, Russia) (Figure 5). The nanomembrane has pores with a diameter of 30–50 nm and it can filter molecules with weights less than 40 kDa. The apparatus has a filling volume of up to 70 mL, and also has the advantage of a single-needle access to a peripheral vein [51].

The characteristics of nanomembrane filter ROSA are as follows [30,40,52,53]:➢ Weight 120 g, size 84 × 84 × 35 mm➢ Amount of blood for venting the membrane is 15–20 mL➢ Separation rate of plasma is 15 mL/min at a speed of blood flow of 70 mL/min➢ Effective surface area 0.15 m^2^ and maximum working pressure up to 40 kPa (300 mmHg)➢ Stable filtration process, thanks to the rigid protective filter coating➢ Reduction of trauma to erythrocytes and other plasma elements➢ Fulfills the requirement of the European Pharmacopoeia Commission and the Committee for Public Health of the European Council (EC Certificate No CQ102011-II) ➢ Reduced priming volume➢ Reduced amount of plasma removed➢ Lower infection and allergic risks➢ Insignificant logistical requirement for transport➢ Fast operation

The process starts with a single peripheral intravenous line, inserted into the patient’s arm. Blood drawn from the patient is mixed with a solution (either crystalloid or colloid solution) and an anticoagulant, most commonly citrate (ACD-A). The pump controls the extracorporeal circulation and conducts the blood through the nanomembrane filter. When the blood fills the first chamber, it passes through the membrane, and the filtration process starts. The blood is then pumped into the plasma chamber, inside the filter, where the corpuscular elements are separated. This system supports the stability of extracorporeal circulation and prevents hemolysis. The filtered plasma is collected in a plastic bag, while the reconstituted blood is returned to the patient by the peripheral intravenous access where the process started (Figure 6) [40,53].

The most frequently used replacement fluid in nanomembrane-based TPE is normal saline (NaCl 0.9), which has the lowest costs and no adverse effects, even when 25% (approximately 700–750 mL plasma) of the circulated plasma is separated [31]. However, replacing plasma with a crystalloid, as a cost-containment strategy, carries a risk of hypotension if the proportion of replacement with the crystalloid exceeds 30% of the circulated plasma [47]. In this setting, significant fluid shifts can occur as water follows its concentration gradient from the intravascular compartment into the extravascular compartment. In our practice of acute TPE treatment with removing of a large volume of plasma (1 to 1.5 of TPV), when a crystalloid was used as a portion of the replacement fluids (a combination of crystalloid and colloid), we administer the crystalloid at the beginning of the exchange and not at the end to avoid significant fluid shifts and hypotension. Our practice of normal saline replacement in the removal of 700–750 mL plasma is in agreement with the so-called LVPE which ranges from 350 mL to 2 L plasma volume removal per procedure. The LVPE is the preferred maintenance TPE treatment modality in chronic conditions, in which the removal of smaller volumes of plasma would be justified for long periods [8]. The relevance of minimizing the adverse effects of the colloid replacement by lowering plasma volume exchanged per treatment (0.5–0.7 of TPV) is supported by the German practice in the field of LVPE as well [7,54]. Their data suggest that effectiveness may be given with volumes below currently recommended volumes (1 to 1.5 of TPV) [4,7]. According to the Spanish practice, the LVPE approach suggests some degree of effectiveness in neuro-immunological disorders (GBS, NMOSD, MG, MS). However, further studies are needed to confirm LVPE as an alternative to classical TPE [8]. Our experience in LVPE contributes by presenting new and up-to-date data on the effectiveness of the low-volume approach after implementing a novel nanomembrane-based technology.

A recent monocentric RCT comparing IA with TPE in patients with neurological autoimmune diseases, who received either tryptophan IA or TPE, showed a marked decrease of the pro-inflammatory cytokines during IA treatment which was in contrast with the tendency of their increase due to the therapy with TPE. One likely reason for the lack of cytokine diminution during TPE treatment could be an eventual activation of cytokine synthesis by the foreign FFP or albumin colloid replacement [55]. Thus, if this hypothesis were to be confirmed, it could shift the practices (depending on their availability) either towards the more efficient IA option or towards the safer nanomembrane-based TPE option, where the physiological normal saline replacement with no adverse effects on the immune system could be an advantage in this context.

## 5. TPE in Guillain-Barre Syndrome

Guillain-Barre Syndrome (GBS) is reported to be caused by antibodies to gangliosides and other neural antigens that cause peripheral nerve damage [56]. Generally, electrophysiology separates demyelinating forms, categorized as “acute inflammatory demyelinating polyradiculoneuropathy” (AIDP), or axonal forms, which have been referred to as “acute motor axonal neuropathy (AMAN), “acute motor and sensory axonal neuropathy (AMSAN) and “acute motor conduction block neuropathy (AMCBN) [57]. Currently, there are no practical treatment differences between the various GBS subtypes, for which only TPE and intravenous immunoglobulin (IVIG) have been observed to be beneficial [58]. TPE, by rapidly removing autoantibodies, helps to overcome the axonal conductivity blocks allowing the nerves to regenerate [59]. As well, eventually leads to a rapid resolution of the symptoms. AAN and ASFA guidelines support using TPE in GBS as an established and effective treatment with a strong level of evidence (Category I: first-line therapy; Grade 1A: strong recommendation, high-quality evidence) [4,60]. TPE is recommended for non-ambulatory patients presenting within 4 weeks and ambulatory patients presenting within 2 weeks [60]. A typical TPE strategy is to exchange 1–1.5 TPV 5–6 times over 10–14 days [4]. Since autonomic dysfunction may be presented, affected patients are potentially more susceptible to volume shifts and hemodynamic instability (hypotension, dysrhythmia) during extracorporeal treatment. The latest research confirmed that GBS patients were more prone to complications than MG patients, mainly hypotensive/vasovagal reactions [61]. These patients are the ideal candidates for our mini-invasive, hemodynamics-friendly nanomembrane-based TPE, which has shown its effectiveness also with the LVPE mode of administration (Figure 7). We did nanomembrane-based TPE early enough in our ICU patients with GBS and they improved without needing invasive respiratory support or early tracheotomy [62]. After six TPE procedures, the patients were discharged afebrile, contact, with stable hemodynamics [50].

## 6. TPE in Chronic Inflammatory Demyelinating Polyradiculoneuropathy

Chronic Inflammatory Demyelinating Polyradiculoneuropathy (CIDP) is an autoimmune disorder involving both T cell-mediated and humoral immune mechanisms by targeting myelin components of peripheral nerves [63]. CIDP can be monophasic, relapsing, or progressive, developing over eight weeks [64]. The presence of autoantibodies against various proteins and glycolipids of the peripheral nerves in samples of serum and CSF from patients with CIPD may provide a rationale for using TPE. Treatment usually consists of either corticosteroids, IVIG, or TPE, followed by long-term immunosuppression. A Cochrane review revealed that there is moderate-to-high-quality evidence that TPE provides a significant short-term improvement in disability, clinical symptoms, and motor nerve conduction velocity recovery in CIDP. However, TPE may be followed by rapid deterioration, with two-thirds of patients suffering a relapse requiring maintenance TPE [65]. The AAN and ASFA concluded that TPE is effective in CIDP and can be offered as a first-line option (Category I), where indicated, with a strong level of evidence (Grade 1B: strong recommendation, moderate quality evidence) [34,61]. A typical TPE strategy is to exchange 1–1.5 TPV over 2 to 3 weeks until improvement followed by maintenance TPE weekly to monthly to control symptoms [4]. Our results in patients with CIDP after implementing the nanomembrane-based TPE (LVPE mode) were inconsistent as well. After a course of five LVPE procedures (Figure 8), the patient had an improvement in his muscle strength and numbness in the upper extremities followed by a relapse 1 month later requiring maintenance TPE or other immunomodulatory therapies [50]. Nevertheless, the plasma exchange in CIDP was positively accepted by our local medical community.

## 7. TPE in Neuromyelitis Optica Spectrum Disorder

Neuromyelitis Optica Spectrum Disorder (NMOSD) is a CNS disorder that predominantly affects the optic nerves (optic neuritis) and the spinal cord (myelitis). It is mediated by aquapoirin-4 IgG autoantibodies (AQP4-IgG) in 70% of patients. A proportion of patients with similar presentation have myelin oligodendrocyte glycoprotein IgG autoantibodies (MOG-IgG) which are also likely pathogenic [66]. Acute attacks are managed by high-dose intravenous steroid treatment and if it fails to resolve the symptoms TPE is applied. TPE removes the pathologic antibodies, immune complex and inflammatory mediators. Plasma exchange improves the short-term prognosis of NMOSD if given early and has proven effective regardless of NMOSD-IgG status [67]. TPE can be helpful in recovery from acute attack but does not prevent further relapses. The AAN recommended TPE in the treatment of fulminant CNS demyelinating diseases (including NMOSD) in case of steroid resistance [60]. The AAN and ASFA conclude that TPE is probably effective in corticosteroid-refractory acute attacks but probably ineffective as maintenance therapy (Category II, second-line treatment; strong level of evidence, Grade 1B: strong recommendation, moderate quality evidence) [34,60]. The timing of TPE to treat acute severe NMOSD is important. The probability of complete recovery reduced from 50% if TPE was given at day 0, to 1−5% if it was delayed to day 20 [68]. Another study from the German Neuromyelitis Optica Study Group found that the use of TPE/IA as a first-line therapy within 3 days of the onset of the attack was associated with good outcomes (40% of patients who received TPE within 3 days returned to baseline compared with less than 4% of those who started TPE/IA after 7 days). The authors suggest that the early start of TPE is strongly recommended [69]. The logistics and availability of TPE do matter [69]. Indeed, the immediate access in our neurological ICU setting to the novel nanomembrane--based TPE technology was in line with the German suggestion of implementing TPE “as early as possible”. In addition, our LVPE approach is in agreement with its applicability in NMOSD patients reported by others as well [5,11]. We present a case of a first-line nanomembrane-based TPE treatment in a patient with a fulminant form of NMOSD. The patient was admitted to NeuroICU with severe neurological deficits (Kurtzke’s EDSS 9 points) accompanied by initial respiratory failure. A total of five TPE procedures were performed (Figure 9). After the third TPE, movements appeared in the right arm and right leg. The initial respiratory failure was controlled. The patient tolerated the procedures well with temporary stabilization and regression of the disease. Unfortunately, during the same in-hospital stay, a subsequent acute attack caused by a new plaque in the fossa rhomboidea of the brainstem led to a fatal cardiac arrest [40].

A second-line nanomembrane-based TPE treatment in another patient with a non-fulminant form of NMOSD with an impressive short-term clinical improvement will be presented below when considering the role of nanomembrane-based TPE in reducing oxidative stress.

## 8. TPE in Myasthenia Gravis

Myasthenia gravis (MG) is an autoimmune disorder caused by the failure of neuromuscular transmission. It is clinically presented by fluctuating muscle weakness and fatigability. A number of circulating autoantibodies against the postsynaptic receptors and related molecules in the neuromuscular junction (NMJ) are involved. They encompass autoantibodies against nicotinic receptor for acetylcholine (AchR) in 80% of cases, against muscle-specific kinase (MuSK) in 10% of cases (often more severe MG), against lipoprotein-related protein 4 (LPR4) in 1–3% of cases (less severe MG. Such antibodies can be observed in 75–95% of MG patients. In antibody-negative cases (comprising 10–15%), neurophysiological tests and the specific response to therapy provide a precise diagnosis [4]. In 15–20% of the cases, patients experience myasthenic crisis (MC), which is a life-threatening condition involving a rapid worsening of the disease and a possible airway compromise from respiratory or bulbar dysfunction, which often requires invasive or non-invasive ventilation. The mechanism of action of TPE in MG includes immediate intravascular reduction of autoantibody concentration, pulsed induction of antibody redistribution, and subsequent immunomodulatory changes. [4]. The removal of circulating autoantibodies restores impaired neuromuscular signal transmission as well as the argin-LRP4-MuSK protein complex binding, which is essential for the formation, maintenance, and regeneration of the NMJ (including the distribution, and clustering of AchR) [70,71]. TPE is indicated in patients with MC such as in patients with respiratory insufficiency or severe dysphagia as well as in patients without MC but with severe MG symptoms, the so-called acute exacerbations of MG, which if left untreated could cause or threaten to cause MC. It is useful in stabilizing the clinical state before thymectomy [72] or in the treatment of refractory MG cases as well. In an acute ICU setting, the prognosis of MC depends on timely therapy with higher mortality when TPE is delayed. Likewise, an early initial trial of NIV may reduce the duration of IMV and may reduce the need for early tracheotomy, as the recovery from MC can be rapid [62]. 

The AAN guidelines state that, because of the lack of RCTs with masked outcomes, there is insufficient evidence to support or refute the efficacy of TPE in the MC or MG pre-thymectomy clinical scenarios (level U = unproven for both indications) [60]. The relative rarity of this autoimmune disease and the unequivocal consensus among clinicians about the efficacy of TPE in MG acute treatment may be the reason why RCTs with the above stringent criteria have not been conducted [66]. However, in the management of moderate to severe MG, including MC, there are two RCTs comparing TPE with IVIG. The authors came to the conclusion that IVIG and TPE are equally effective [73,74]. Moreover, treatment with TPE led to improved clinical outcomes in MG patients not responsive to IVIG, as shown in a retrospective review [75]. Unlike the AAN statement, the ASFA recommends TPE as an effective acute short-term treatment for moderate-severe MG including MC, as well as for unstable or refractory MG cases, including unstable disease activity pre-thymectomy (Category I, first-line option; strong level of evidence—Grade 1B: strong recommendation, moderate quality evidence) [4]. A typical TPE cycle consists of 3–6 treatments with 1–1.5 TPV exchange over 2 weeks, but an LVPE mode of treatment with less than 1 TPV exchange can be beneficial as well [4,6,76]. 

In a recent before-and-after single-center observational study we revealed the efficacy of the nanomembrane-based TPE in treating patients with MG acute exacerbation (including MC) compared to the conventional treatment [51]. TPE or IVIG infusions were used in impending/manifested MC, especially in patients at high-risk for prolonged invasive mechanical ventilation (IMV) and in those tolerating non-invasive ventilation (NIV). The clinical improvement was estimated using the Myasthenia Muscle Score (MMS) (0–100), with ≥20 increase for responders. The basic outcome measures involved the rates of TPE, IVIG, and corticosteroids therapies, NIV/IMV, early tracheotomy, MMS scores, extubation time, neuro-ICU/hospital LOS, complications, and mortality rates. The univariate analysis showed that IMV was lower in the nanomembrane-based group (42%) compared to the conventional treatment group (83%) (*p* = 0.02) (Table 3). The multivariate analysis using binary logistic regression showed that the nanomembrane-based TPE and NIV served as independent predictors for short-term (≤7 days) respiratory support (*p* = 0.014 for TPE; *p* = 0.002 for NIV). Age also occurred as a factor that adversely affected the responder rates in a time-dependent fashion (Table 4). Patients with early-onset MG (<50 years) showed a better response to the applied treatment than those with late-onset MG (>50 years) (Figure 10). The TPE technology moved our clinical practice towards proactive (in terms of earlier implementation of nanomembrane-based TPE) rather than protective treatment in reducing prolonged IMV during MG acute exacerbations [51]. 

In addition, the majority of our patients received the LVPE mode of TPE (Figure 11) which corroborates its usefulness in MG treatment reported by others [4,6,76]. In another study, the nanomembrane-based TPE was used to treat acute respiratory distress syndrome (ARDS) in a 41-year-old man with a history of MG in remission, who had developed ARDS secondary to pneumonia. When his oxygenation continued to deteriorate irrespective of IMV, the nanomembrane-based TPE was employed, and after three sessions, the patient was successfully weaned off the ventilator [29]. Based on the Bulgarian experience we strongly recommend the nanomembrane-based TPE as an effective early treatment in MG acute exacerbations, especially in patients requiring respiratory support. More importantly, these two publications [29,51] meet the ASFA’s Writing Committee criteria of a minimum of 10 cases published in the last decade in peer-reviewed journals, ideally by more than one research group, for the nanomembrane-based TPE to be included as a new treatment option in the next ASFA guidelines MG fact sheet [77]. 

## 9. TPE in Multiple Sclerosis

Multiple sclerosis (MS) is an autoimmune multifocal CNS inflammatory disease, characterized by chronic inflammation, demyelination, axonal damage, and subsequent gliosis. The most affected regions by MS in the CNS are the periventricular area, subcortical area, optic nerve, spinal cord, brainstem, and cerebellum. MS is categorized as remitting relapse (RR), secondary progressive (SP), and primary progressive (PP) [78,79]. Despite the specific clinical form, the initial stage of the disease is accompanied by an exacerbated immune and inflammatory response, whereas the later stages are characterized by decreased inflammation, marked neurodegeneration, and significant disability [80,81]. The pathophysiology of MS suggests that in genetically susceptible subjects, independent populations of CD4+ T lymphocytes are activated in the immune system, migrate across the blood-brain barrier, and trigger CNS tissue damage. They release inflammatory cytokines, initiate cytotoxic activities of microglia with the release of nitrous oxide and other superoxide radicals, stimulate B cells and macrophages, and activate the complement system [82]. Autoantibodies against myelin basic protein and myelin oligodendrocytes glycoprotein have been detected in patients with MS. These antibodies may mediate injury by complement fixation or linking with innate effector cells such as macrophages [82]. As in any other immune-mediated neurological disorder with acute severe deterioration, MS patients need a rapid intravascular reduction of the autoantibodies, complement complexes, pro-inflammatory cytokines, and reactive oxygen species, which is the clinical rationale for the application of TPE. 

The most common symptoms of MS include spasticity, chronic pain, fatigue, motor and mobility disorders, and cognitive impairment [83]. Acute attacks of MS are manifested with an acute, severe neurological deficit such as coma, aphasia, acute severe cognitive dysfunction, hemiplegia, paraplegia, or quadriplegia [60]. Relapses of MS are manifested by the reappearance of clinical symptoms such as acute demyelinating optic neuritis, diplopia, limb weakness, gait ataxia, neurogenic bladder and bowel symptoms, etc. [30]. The standard treatment of acute MS attacks or relapses consists of immunosuppression with intravenous administration of high-dose corticosteroids. The aim of a relapse treatment is to accelerate functional recovery after inflammatory demyelination, alleviate the severity of the relapse and decrease the chance of persistent neurologic deficit [66]. If patients are unresponsive, which occurs in 20–25% of cases, after an interval of 10–14 days a second corticosteroid pulse therapy in combination with TPE is recommended. TPE in steroid-refractory acute attacks/relapses is recommended as an adjunctive treatment by the AAN (Level B recommendation) [60] and as a second-line treatment by the ASFA (Category II; Grade 1B: strong recommendation, moderate quality evidence) [4]. In this acute clinical scenario, a course of 5–7 TPE procedures with 1–1,5 TPV exchange over 2 weeks have a response rate of more than 50% [4]. In contrast, according to the AAN evidence-based guideline the TPE should not be offered for chronic PP or SP forms of MS (Level A recommendation) [60]. Interestingly, a recent retrospective two-center study revealed a 50% response rate for the PP/SP subgroup of patients with MS, treated with TPE/IA (both IA and TPE were equally effective) [84]. This observation implies that TPE could be considered an escalation therapy in progressive MS as well [66]. 

We carried out a second-line nanomembrane-based TPE in steroid-refractory MS in 15 patients with RR form of MS [31,32,40,49] and in one patient with progressive MS [49]. The Bulgarian short-term therapeutic algorithm included four sessions of nanomembrane-based TPE with LVPE mode, 0.8 TPV exchange (Figure 12), implemented every other day, followed by 5th TPE after 1 month, 6th TPE 3 months later, and 7th TPE 6 months later [9,40]. After implementing a cycle of 4 TPE usually, the symptoms of ocular and vestibular motor function, of visual acuity, of walking without assistance, etc., as well as those of acute neurological deficit (Kurtzke’s EDSS) were improved significantly. A remarkable reduction of the markers of oxidative stress (addressed in more detail below) was observed as well. We suggest our LVPE treatment with the nanomembrane-based TPE in patients with acute, severe neurologic deficits caused by MS, who have a poor response to treatment with high-dose corticosteroids. 

## 10. The Role of Nanomembrane-Based TPE in Reducing Oxidative Stress

Oxidative stress (OS) is a known pathogenetic factor in the onset and development of a significant proportion of neurodegenerative diseases, including MS, Alzheimer‘s disease, Parkinson‘s disease, and others. These neurodegenerative diseases, characterized by a variety of symptoms and clinical manifestations, are almost always associated with inflammatory processes and varying degrees of impaired motor activity in patients. It is known that inflammation increases the level of free radicals in the body, leading to a state of OS [50]. OS is essentially an imbalance in oxidative and antioxidative status which can cause neuroinflammation and neurodegeneration [85]. As OS could be implicated in NMOSD, an antioxidant approach may be of interest [86]. Given the fundamental role of AQP4 in the pathogenesis of NMOSD decreasing AQP4 activity could be of interest as well [87]. In animal models of transient cerebral ischemia, the upregulation of AQP4 is at least in part dependent on OS generated by the ischemia. The administration of antioxidant drugs before and two times after the ischemic procedure in rats resulted in reduced infarct size and brain edema along with an augmentation of antioxidant systems and a reduction of AQP4 overexpression [88]. This preclinical evidence provides one more insight into the pathophysiological rationale for reducing OS in this neuroinflammatory and neurodegenerative disorder. OS is suggested as a major pathogenic factor in the etiology of MS as well [89]. 

The brain is highly susceptible to OS due to high oxygen demands in its metabolic pathways, abundant content of more easily peroxidizable polyunsaturated fatty acids (PUFAs), and low activity and quantity of antioxidant enzymes compared to other tissues [90]. Reactive oxygen species (ROS) and reactive nitrogen species (RNS) are reported to play a significant role in disease onset and progression by participating in myelin and oligodendroglia degeneration, the latter serving as pathophysiological hallmarks of MS [91]. There are numerous reports in the literature, implying that relapsing-remitting and progressive clinical phases of MS are associated with distinct mechanisms, for example, focal inflammation is related to the onset of relapses, whereas axonal degeneration may underlie disease progression. OS in MS is suggested to be related to both processes [90,92]. Myelin, which surrounds the axons and is the target of immune attacks in MS, consists of 30% protein and 70% lipids. Malondialdehyde (MDA) is a highly reactive organic compound that is produced at the lipid peroxidation of PUFAs. The presence of this aldehyde in the body is used as a biomarker to measure the level of OS [50]. ROS has a multifaceted role in the pathogenesis of MS lesions [93]. Data suggest that ROS are accumulated as a result of interaction between monocytes and the brain endothelium, which induces tight-junction alterations, cytoskeleton reorganization, and disturbance of the blood-brain barrier integrity, subsequently resulting in extravasation of leukocytes into the CNS [93,94]. What is more, the infiltrated leukocytes produce additional amounts of ROS, which cause myelin phagocytosis [95], destruction of the oligodendroglia [96], and subsequent neuronal and axonal damage [97].

If looked at from a broader perspective, a significant part of the metabolism of brain structures is affected by OS. Data revealed the presence of extensive oxidative damage to proteins, lipids, and nucleotides [98,99]. Studies performed in the CSF have also shown increased levels of lipid peroxidation markers such as MDA [100], and isoprostanes [100]. The latter are derivatives of arachidonic fatty acid, which is from the group of omega-6 fatty acids and whose four double bonds are highly susceptible to oxidative destruction. Quantification of isoprostanes is proposed as an indicator of free radical generation and is considered a highly reliable marker of the levels of total OS [50].

In some of our NMOSD and MS patients, we used a complex approach to assess the level of OS before and after TPE by examining three of the well-established markers of the degree of OS, namely MDA (µg/L), ROS (FI/mg protein) and isoprostanes (pg/mL).

A patient with NMOSD with failed first-line treatment (corticosteroids with azathioprine) underwent a second-line treatment with nanomembrane-based TPE. After the completion of four LVPE mode nanomembrane-based TPE procedures, the patient had an impressive short-term clinical improvement (from being confined to a wheelchair for one year to being able to walk again without assistance after the end of the TPE course of treatment). The indicators of OS (MDA, ROS, and isoprostanes were examined before and after TPE during the course of four procedures. The obtained results showed that TPE was accompanied by a decrease in the level of all three analyzed parameters that are indicative of OS (Figure 13, Figure 14 and Figure 15). More importantly, the reduction in OS was accompanied by the apparent clinical improvement which was assessed according to the international scale of Kurtzke EDSS (Kurtzke’s EDSS before TPE 8 points, after TPE 4 points) [50]. The effect of TPE on ophtalmological status examined by optical coherence tomography scanning is presented in Table 5 [48].

In a case series of five patients with RR form of MS with failed first-line treatment, an escalation to a second-line treatment with nanomembrane-based TPE was implemented [48,50]. Most of the patients received three LVPE mode nanomembrane-based TPE procedures (patients 1, 2, 3, 5), whereas one of them received four LVPE mode nanomembrane-based TPE procedures (patient 4). Monitoring of OS markers showed that carrying out TPE leads to a decrease in the level of OS (Figure 16), which was favorable for the patients’ general and neurological status. TPE, especially in cases with a lower rate of disability (Kurtzke EDSS ≤ 5 points for RR form of MS [85], i.e., with a predominant inflammatory component in the occurred relapse rather than a neurodegenerative one), has been found to have a relatively beneficial effect on the clinical symptoms, resulting in overall stabilization over the course of the disease. For establishing the responders’ rate a clinically meaningful reduction of 0.5 points on the Kurtzke EDSS was adopted [7]. Only in one patient (patient 3) there was no improvement in the assessed clinical disability (Kurtzke EDSS before TPE 8.5 points, after TPE 8.5 points). In these case series focused research on the level of OS before and after TPE [48,50], the observed 80% clinical improvement (according to Kurtzke EDSS, Table 6) was consistent with a 74% response rate in the same subgroup of patients (RR form of MS) reported by others [84].

Our findings suggest that the LVPE mode nanomembrane-based TPE is an effective second-line treatment in neuroinflammatory and neurodegenerative diseases not only as an immunomodulatory approach in decreasing the pathogenic large molecules (autoantibodies, immune complexes, cytokines, etc.), but as a therapeutic approach in reducing OS as well. How this would translate into neuroregeneration in CNS autoimmune attacks (similar to the biogenic selenium nanoparticles therapeutic antioxidant, anti-inflammatory, and neuroregenerative activities in rats with spinal cord injury [101]) is a hypothesis that should be confirmed or rejected in future studies. The observed rapid recovery of peripheral nerves’ electrical conductivity along with the improvements in symptoms and disability in GBS and CIDP suggest such neuroregenerative TPE properties as well.

## 11. Cost-Effectiveness Considerations

Immune-mediated neurological disorders are often treated initially with corticosteroids followed by an escalation therapy with TPE or IVIG when not responding to the corticosteroid treatment. The efficacy of TPE and IVIG in most cases is recognized to be comparable [102]. The choice of TPE is usually influenced by evidence-based medicine: as a first-line treatment in conditions with established effectiveness to TPE (such as GBS, CIDP, MG) or as a second-line treatment in conditions in which the effectiveness of TPE is less convincing (such as NMOSD, MS). Because TPE and IVIG treat similar conditions, after a review of the efficacy evidence, the attending physician needs to choose which one to use. The decision-making is influenced by the ease of delivery, the side effect profile of each, and last but not least their cost-effectiveness.

The systemic reactions to IVIG infusion are often self-limiting, of mild to moderate severity, and can often be avoided by slowing down the infusion rate. IVIG carries risks of rare, but of clinical significance complications such as renal failure, hypercoagulable states, and hypersensitivity to immunoglobulin for IVIG infusion. TPE is relatively safe with the most significant side effect associated with the method of vascular access. Central venous access carries a higher risk of complications than peripheral access [103]. The risk could be reduced by implementing the centrifugal TPE or the nanomembrane-based TPE via peripheral access. The same applies to TPE complications such as hemodynamic instability and hypersensitivity to albumin volume replacement which could be reduced again by applying the nanomembrane-based TPE [31]. These side effects and their treatment are an additional financial burden that always comes into consideration.

Cost-minimization analyses showed that TPE was less costly than IVIG for GBS in the USA and UK, whereas in Taiwan IVIG treatment for GBS was more cost-effective than TPE because of shorter hospital stays and lower in-hospital costs [104,105,106]. Even in countries with low-GDP (such as India, Turkey, and Brazil), TPE is a less expensive alternative for GBS treatment [107,108,109]. A retrospective study of insurance claims records revealed that TPE alone or in combination with corticosteroids was less costly than IVIG alone or in combination with corticosteroids for CIDP [110]. In addition, a Cochrane Systematic Review on CIDP concluded that, although costly TPE is still less expensive than IVIG in most countries and should be considered as an alternative treatment [111]. Two different cost comparison studies in MG acute treatment setting revealed that from a university hospital perspective, IVIG was less expensive than TPE, whereas in MG chronic treatment setting the cost-minimization evaluation was in favor of TPE over IVIG [112,113,114].

The largest-ever cost-analysis of TPE versus IVIG in the treatment of selected immune-mediated neuromuscular disorders (GBS, CIDP, MG acute, and MG chronic) was conducted in the US outpatient and inpatient hospital settings [102]. The analysis of outpatient costs revealed that mean total hospital costs in the TPE group were significantly lower compare to the IVIG group (*p* < 0.0001). After adjusting for confounding variables the cost breakdown favored TPE again with the highest average hospital costs for MG acute treatment, followed by GBS, CIDP, and MG chronic treatment. Similar cost patterns were identified in total therapy costs. The analysis of inpatient costs showed that although not significant, the unadjusted total hospital costs were higher for TPE compared to IVIG in patients with GBS, MG chronic, and MG acute treatments, whereas these costs were lower for TPE compared to IVIG in patients with CIDP. After adjusting for confounding variables the trend continued in the same direction with the exception of significantly increased costs for MG acute treatment. Exactly the opposite trend was observed for inpatient unadjusted and adjusted total therapy costs where the costs of TPE were lower compared to IVIG across all cohorts (*p* < 0.0001). In financial terms, this represents over 65% savings for therapy costs when TPE was used instead of IVIG for inpatient admissions [102]. However, these data should be interpreted with caution in the context of the local healthcare system. In the US settings, older patients are more likely to receive IVIG than TPE, because up to 73% of IVIG treatment is covered by Medicare [102]. Thus elderly patients could incur more total hospital costs given their illness severity, comorbidities, more intensive care, etc. Nevertheless, in the NHS settings, the latest UK cost-minimization analysis of TPE versus IVIG in the treatment of autoimmune neurological conditions confirmed that a day-case TPE service, delivered mainly using peripheral venous access, was able to treat patients previously stabilized on IVIG with comparable clinical outcomes and at much lower cost [103].

The Hemophenix apparatus with ROSA nanomembrane is easily adaptable to outpatient settings (Figure 4). Their insignificant logistical requirement for transport, fast operation, and LVPE mode of action with normal saline volume replacement by using peripheral venous access are undeniable advantages in both outpatient and inpatient settings. In addition, the lower average costs of implementing Hemophenix Nanofiltration TPE (Table 7) could make the whole process of a day case service a viable cost-effective alternative to the conventional TPE or IA modalities [52]. Regardless of the described current and potential future benefits, the same limitations that apply to the US healthcare setting apply to our local one. Our novel nanomembrane-based TPE modality should be interpreted with caution and should be placed in the context of local peculiarities concerning our study population, local experience, availability, and insurance coverage [51].

## 12. Summary and Conclusions

In summary, the main drawback of centrifugation-based TPE mode is related to strong mechanical shear stress inducing platelet activation and causing hemolysis which could reduce its efficacy and safety [22]. On the other hand, the membrane-based TPE mode has inherent limitations because of low selectivity, especially towards components of the same size, which could decrease its usefulness as well [22]. With the advent of nanotechnologies, new perspectives for reducing the amount of membrane filtration area are discovered. The innovative biomaterials such as polysulfone, polyarylethersulfon, polyacrylonnitrile, silicon nitride, etc., combined with the advanced nanospinning technology create new synthetic ultrathin dialysis membranes with much higher performances and lower bioreactivity profile [24,115,116,117,118,119,120,121,122,123,124,125,126,127,128]. However, these highly efficient membranes are needed reduced extracorporeal volume, on-device anticoagulation, and reliable peripheral vascular assess in order to pave the way for continuous wearable dialysis therapy [116]. Our novel nanomembrane-based TPE technology is largely a successful continuation in the real clinical practice of this revolutionary change requiring a portable device, a small size nanomembrane, a small volume of the extracorporeal circuit, and low filtration volumes. It still has a long way to go from the current investigational administration to the future implementation on a large scale, which is the main limitation of this LVPE approach.

In addition, the COVID-19 pandemic catalyzed remote management models of patient care which comprised along with traditional hospital-based fixed-site therapeutic apheresis services, mobile apheresis services and on-site point services as well [129]. The ASFA emphasized that the technological capability exists to provide a higher quality of clinical care to patients with rare autoimmune diseases (such as those focused on in this review) remotely [130]. Given the successful removal of viruses during the manufacturing of biopharmaceutical products by means of the nanomembrane filtration process [131,132,133,134,135,136,137,138,139,140,141,142,143,144], the nanofiltration technology in general, and our novel nanomembrane-based LVPE technology in particular, would be extremely useful in the context of future viral pandemic settings, and as one more novel technological capability in the ASFA future recommendations in the field as well.

Nanomembrane-based technology is a new and promising type of TPE, which seems to share the basic advantages of the conventional one, but probably with fewer adverse effects. It could play a valuable role in the patient management by ameliorating neurological symptoms, improving disability, and reducing oxidative stress in a cost-effective way. Further research is needed to identify which patients benefit most from this novel TPE technology.

## Figures and Tables

**Figure 1 biomedicines-11-00328-f001:**
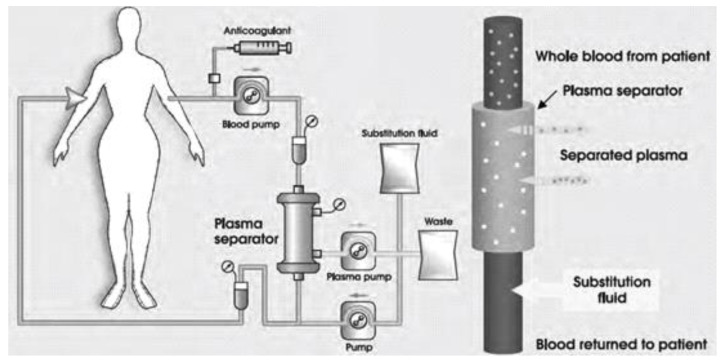
Classic membrane TPE.

**Figure 2 biomedicines-11-00328-f002:**
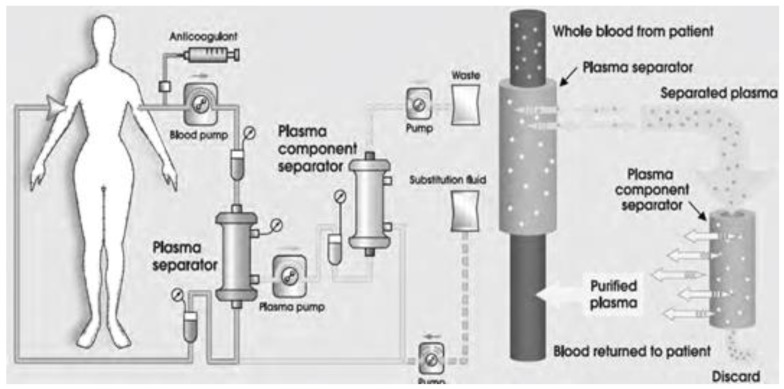
Double filtration (or cascade filtration) plasmapheresis (DFPP).

**Figure 3 biomedicines-11-00328-f003:**
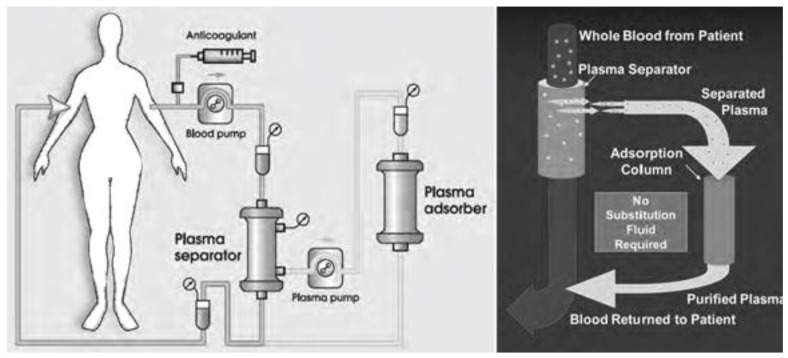
Immuno-adsorption (IA) plasmapheresis.

**Figure 4 biomedicines-11-00328-f004:**
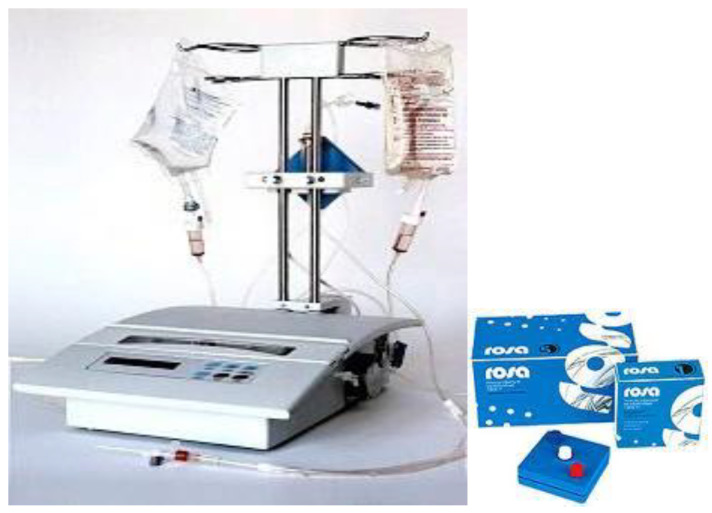
Hemophenix apparatus with ROSA nanomembrane [40,42].

**Figure 5 biomedicines-11-00328-f005:**
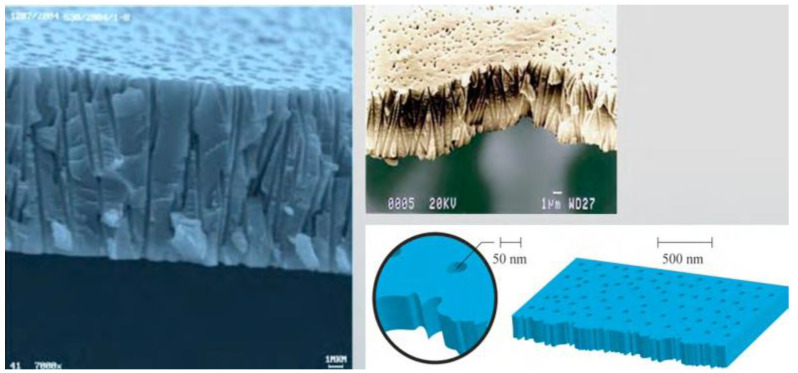
Electron microscopic profile of the track membrane ROSA [40,42].

**Figure 6 biomedicines-11-00328-f006:**
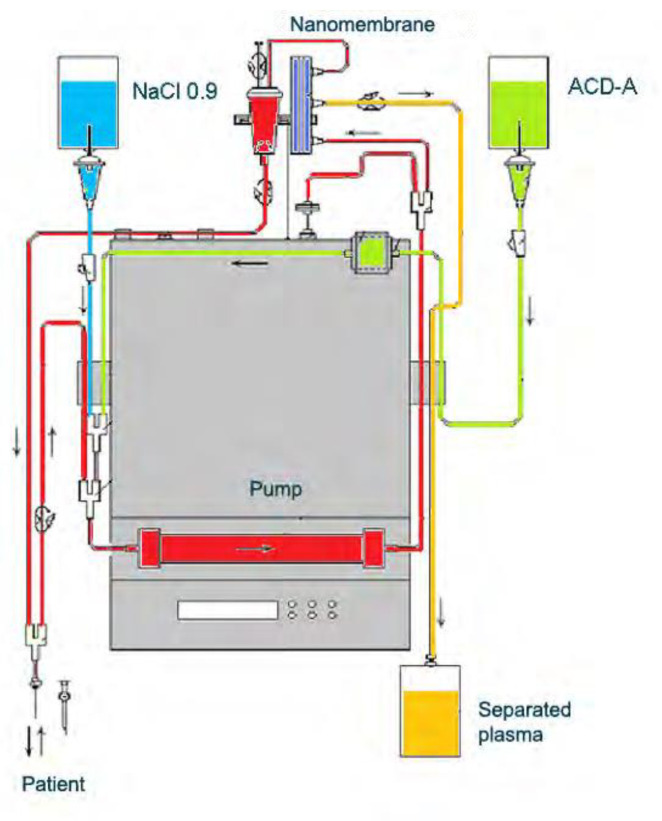
Diagram of nanomembrane-based TPE device [40,42].

**Figure 7 biomedicines-11-00328-f007:**
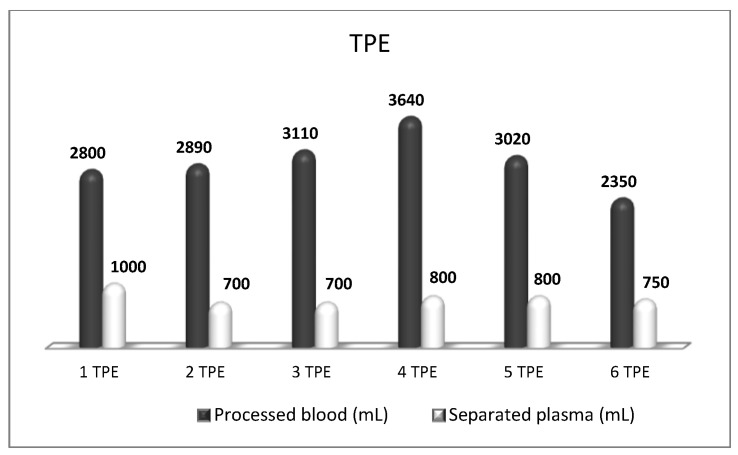
Amount of processed blood and separated plasma (LVPE) during a therapeutic course of a total of six TPE in a patient with GBS.

**Figure 8 biomedicines-11-00328-f008:**
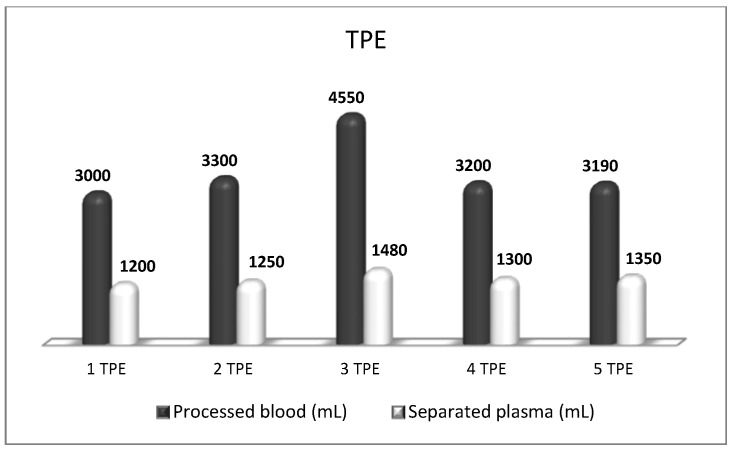
Amount of processed blood and separated plasma (LVPE) during a therapeutic course of a total of five TPE in a patient with CIPD.

**Figure 9 biomedicines-11-00328-f009:**
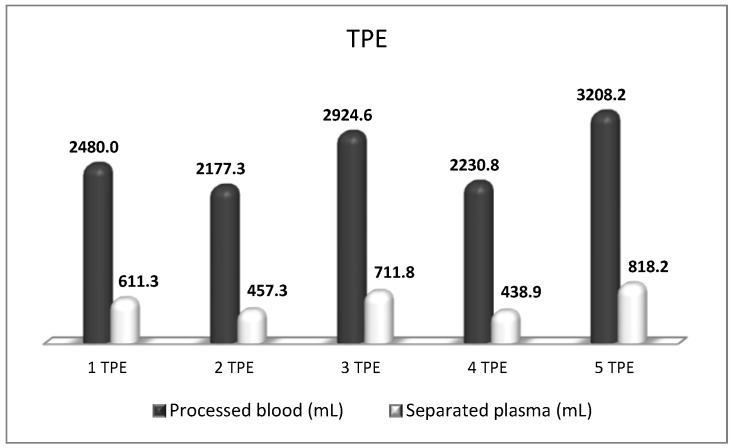
Amount of processed blood and separated plasma (LVPE) during a therapeutic course of a total of five TPE in a patient with a fulminant form of NMOSD (Devic’s disease).

**Figure 10 biomedicines-11-00328-f010:**
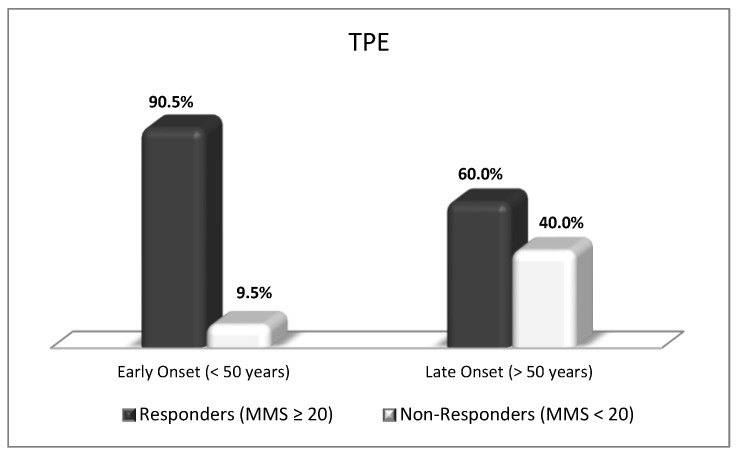
Proportions of responders and non-responders in early-onset (*n* = 21) and late-onset (*n* = 15) MG patients (MMS—Myasthenia Muscle Score) (*p* < 0.05).

**Figure 11 biomedicines-11-00328-f011:**
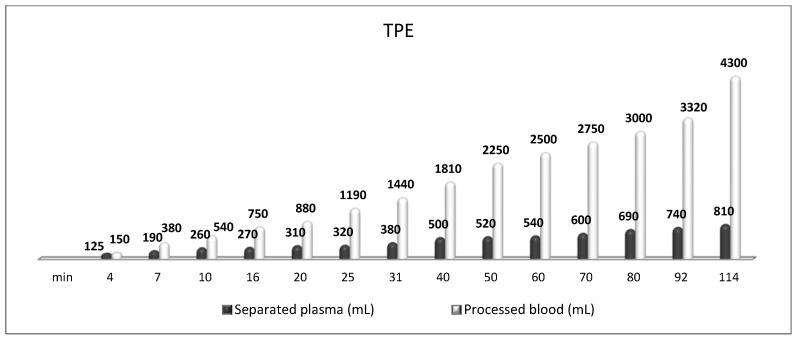
Amount of processed blood and separated plasma (LVPE) during a TPE procedure in a patient with MG.

**Figure 12 biomedicines-11-00328-f012:**
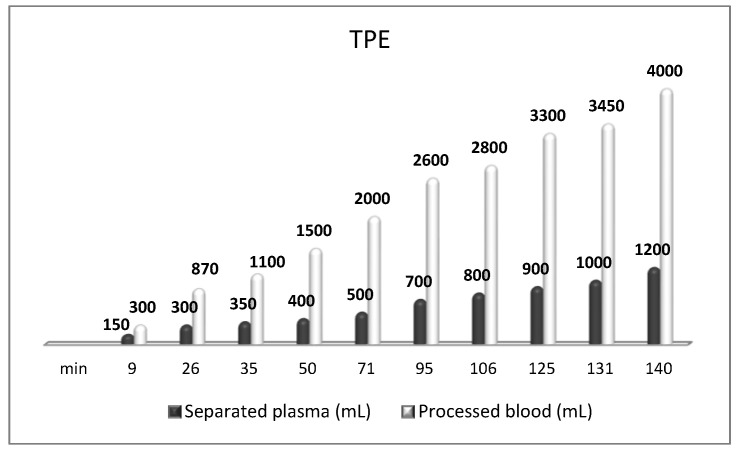
Amount of processed blood and separated plasma (LVPE) during a TPE procedure in a patient with MS.

**Figure 13 biomedicines-11-00328-f013:**
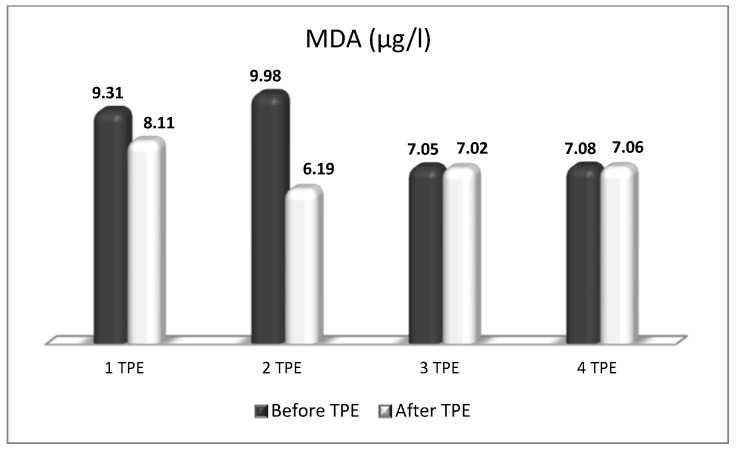
Comparison of malondialdehyde (MDA) values before and after four TPE procedures in a patient with a non-fulminant form of NMOSD with impressive short-term clinical improvement.

**Figure 14 biomedicines-11-00328-f014:**
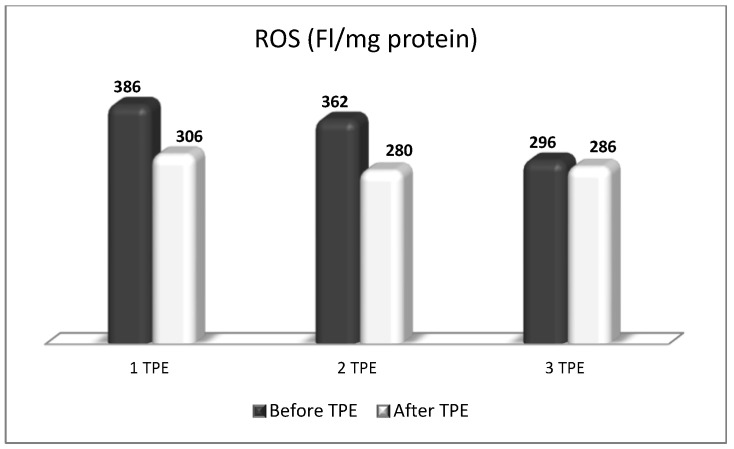
Comparison of reactive oxygen species (ROS) values before and after three TPE procedures in a patient with a non-fulminant form of NMOSD with impressive short-term clinical improvement (FI/mg protein = Fluorescence Intensity/mg protein).

**Figure 15 biomedicines-11-00328-f015:**
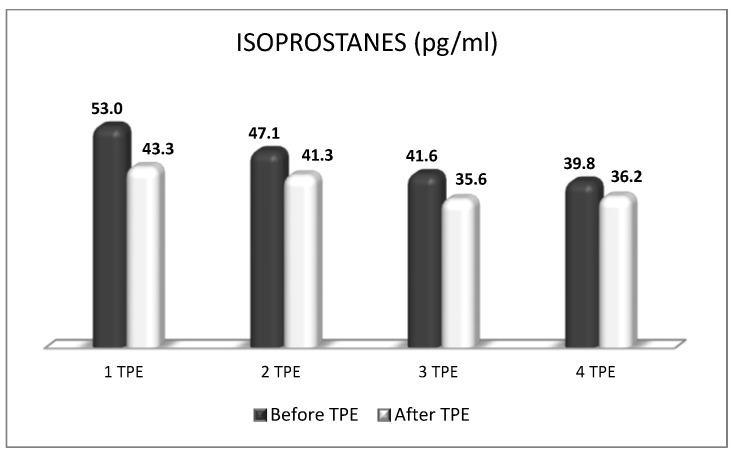
Changes in the level of isoprostanes before and after plasmapheresis in a patient with a non-fulminant form of NMOSD with impressive short-term clinical improvement (*p* < 0.05).

**Figure 16 biomedicines-11-00328-f016:**
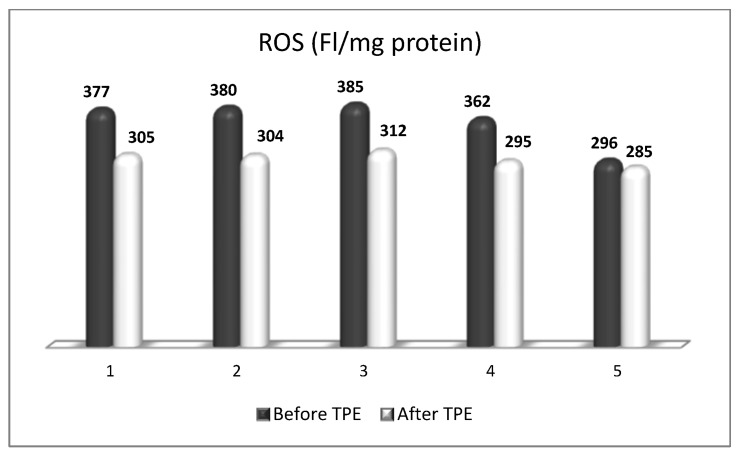
Changes in the ROS content of patients (1, 2, 3, 4, 5) with MS before and after plasmapheresis. (FI/mg Protein = Fluorescence Intensity/mg Protein).

**Table 1 biomedicines-11-00328-t001:** Membrane-based nanotechnologies published within and beyond the scope of the last review on blood apheresis technologies (first published on 18 October 2021).

Type of Nanotechnology	Type of Research	Applications	References
**Covered in the last review on blood apheresis technologies**			[22]
Membrane with silicon dioxide nanoparticles	Experimental	Hemodialysis	[23]
Membrane with ferric oxide nanoparticles	Experimental	Hemodialysis	[24]
Membrane with multi-walled carbon nanotubes	Experimental	Hemodialysis	[25]
Membrane with graphene oxide nanoparticles	Experimental	Hemodialysis	[26,27]
**Not covered in the last review on blood apheresis technologies**			[22]
Membrane with nanoscale pores made of Lavsan film irradiated by accelerated in a collider charged argon particles	Clinical	TPE	[9,28,29,30,31,32,33]

**Table 2 biomedicines-11-00328-t002:** Comparison of the most common TPE techniques.

	Membrane Filtration	Centrifugal Separation
Venous access	Central venous access	Peripheral or Central venous access
Anticoagulation	Heparin	Citrate
Efficiency of plasma removal	30%	70%
Blood flow rate	High (100–150 mL/min)	Low (50–70 mL/min)

**Table 3 biomedicines-11-00328-t003:** Treatment and outcomes before and after the introduction of the novel nanomembrane-based TPE technology in patients with MG acute exacerbations (ICU—intensive care unit).

	Conventional Treatment *n* = 24	Nanomembrane-Based TPE *n* = 12	*p* Value
Non-invasive ventilation	9 (37%)	7 (58%)	0.236
Invasive ventilation	20 (83%)	5 (42%)	0.020
Early tracheotomy	12 (50%)	3 (25%)	0.286
Extubation times (days)	17 ± 21	5 ± 7	0.023
ICU length of stay (days)	20 ± 24	10 ± 5	0.118

**Table 4 biomedicines-11-00328-t004:** Predictors of short-term (≤7 days) respiratory support in patients with MG acute exacerbations (MGFA—Myasthenia Gravis Foundation of America; ICU—intensive care unit).

	OR	95% CI of OR	*p* Value
Age	0.942	0.896	0.018
MGFA class (IV/V) on ICU admission	10.111	2.086–48.999	0.004
Therapeutic plasma exchange	9.000	1.550–52.266	0.014
Non-invasive ventilation	12.000	2.484–57.975	0.002

**Table 5 biomedicines-11-00328-t005:** The effect of TPE on ophtalmological status in a patient with a non-fulminant form of NMOSD with impressive short-term clinical improvement (BCVA—best corrected visual acuity; RNFL—retinal nerve fiber layer thickness; MD—mean deviation of the visual field).

	BCVA		Average RNFL Thickness (µm)		MD (dB)	
	Before TPE	After TPE	Before TPE	After TPE	Before TPE	After TPE
Right eye	1.0	1.0	71.0	59.5	−8.72	−7.81
Left eye	1.0	1.0	65.5	60.25	−15.77	−15.12

**Table 6 biomedicines-11-00328-t006:** The effect of plasmapheresis on oxidative stress (ROS, Isoprostanse) and disability scores (Kurtzke EDSS) in patients with MS (ROS—reactive oxygen species; EDSS—expanded disability status scale).

Patient Number	ROS (FI/mg Protein)		Isoprostanes (pg/mL)		Kurtzke EDSS	
	Before TPE	After TPE	Before TPE	After TPE	Before TPE	After TPE
1.	377	305	56	42	6.5	6.0
2.	380	304	59	48	6.5	6.0
3.	385	312	95	90	8.5	8.5
4.	362	295	61	51	3.0	2.5
5.	296	285	62	54	6.0	5.5

**Table 7 biomedicines-11-00328-t007:** Average costs of certain TPE procedures (adapted from [52]).

	Hardware Cost		Management Cost		
Type of Procedure	Consumables	Replacement Fluids and Medications	Various Costs	Personal Cost	Total Cost
	€	€	€	€	€
Centrifugal TPE	186.58	356.00	50.00	200.00	792.58
Hemophenix Nanofiltration TPE	150.00	15.00	50.00	100.00	315.00
Filtration TPE	175.07	356.00	50.00	200.00	781.07
Cascade filtration TPE	404.89	85.00	50.00	200.00	739.89
Staphylococcal protein A column- immunoadsorption	1275.07	20.00	50.00	200.00	1547.07

## Data Availability

No new data were generated.

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
