# Peer review of "Therapeutic Plasma Exchange in Certain Immune-Mediated Neurological Disorders: Focus on a Novel Nanomembrane-Based Technology"

_biomedicines, 2023, doi:10.3390/biomedicines11020328_

Round 1

Reviewer 1 Report

The work deals with Therapeutic plasma exchange (TPE) method compared to low volume plasma exchange (LVPE) and nanomembrane-based therapies for blood purification. Authors used a commercial ROSA nanomembrane to compare with TPE and LVPE for five most common neurological diseases of Guillain-Barre Syndrome (GBS), Chronic Inflammatory Demyelinating Polyradiculoneuropathy (CIDP), Neuromyelitis Optica Spectrum Disorder (NMOSD), Myasthenia Gravis (MG) and Multiple Sclerosis (MS). The review was well organised and it can be published after some minor changes. 

C1) Figure 11 supposed to be Figure 12, please correct it. 

C2) The conclusion section is weak. Please add the main idea and outcome of the review clearly. 

Author Response

We would like to thank this reviewer for his/her recommendations, which, no doubt, would contribute to making this paper an interesting source of information. We have taken into consideration every point raised in this review and have made the required changes as follows:

1) Reviewer: Figure 11 supposed to be Figure 12, please correct it.

Answer: On line 463 we did the necessary change.

2) Reviewer: The conclusion section is weak. Please add the main idea and outcome of the review clearly.

Answer: We added the necessary information on lines 644-672 accordingly.

Reviewer 2 Report

The review by Dimitar Tonev et al. gives an overview on the role of nanomembrane-based technology as an alternative LVPE treatment option in the neurological conditions. Some major revisions should be performed before publication.

1. The authors are recommended to give the number of papers and patents (a graph could be added) in this field in the introduction in order to justify a review in this topic. The keywords used to make this graph should be added in the figure legend.

2. Some recent reviews in the area should be mentioned in the introduction section. A justification for an additional review in this field should be given.

3. A graphical abstract that can reflect the scope of the review well is recommended to be drawn and presented in this paper, which will be helpful for understanding of readers.

4. Did the author get the copy rights of Figure 1-3? If so, it should be claimed.

5. In the section 3, what types of methods were employed for the fabrication of nanomembranes? Please give a systematic summary for the generation methods of nanomembranes including their merits and demerits?

6. Most recently, Electrospinning has been widely employed to fabricate nanofibrous filtration materials (10.1016/j.progpolymsci.2013.06.002, 10.1007/s10853-014-8308-y and 10.1016/j.apmt.2022.101473). Is it suitable for the fabrication of nanomembranes for TPE application? Please add some more related discussion .

7. Some statements feel they are lacking references. Admittedly, some of these statement might be considered well known facts, the concepts mentioned might have been referenced previously or will be in the future, but it might still be pertinent add references next to these statements for new readers to the field, skim readers or people who don't necessarily want to go looking for the relevant reference.

8. The authors should make a critical review instead of plain text flow. This means that a comparative discussion should take place assisted with categorization of the attributes of the different systems in each section/Tables.  

9. As well known that, nanomembrane-based strategies have been widely investigated for TPE application. Do they have any commercial products available now?

10. The challenging and future direction in this area should be discussed in the conclusion section。

11. The paper contains some typo and graphical errors. Please read carefully and correct Them.

Author Response

We would like to thank this reviewer for the useful remarks, which, no doubt, would improve the quality of this paper and would make it more readable and comprehensive.

We have taken into consideration every point raised in this review and have made the required changes as follows:

1) Reviewer: The authors are recommended to give the number of papers and patents (a graph could be added) in this field in the introduction in order to justify a review in this topic. The keywords used to make this graph should be added in the figure legend.

Answer: As suggested by this Reviewer we have added to the Introduction section new references [10-15] (along with our cited research in the first submission) in a table format in order to justify strongly the need of a new review. Given the experimental type of part of the cited research [11-15] it was impossible to give their number of patients.

2) Reviewer: Some recent reviews in the area should be mentioned in the introduction section. A justification for an additional review in this field should be given.

Answer: We included the last review [10] of relevance discussing the membrane nanotechnologies, accordingly.

3) Reviewer: A graphical abstract that can reflect the scope of the review well is recommended to be drawn and presented in this paper, which will be helpful for understanding of readers.

Answer: We added a graphical abstract in the Abstract section, accordingly.

4) Reviewer: Did the author get the copy rights of Figure 1-3? If so, it should be claimed.

Answer: The author A.M. have the copy rights of Figure 1-3 as a co-author in the two published books (in Bulgarian and in Spanish). Nevertheless, in line with the referee 2 last suggestion and the referee 3 comments we remodeled these graphs in order to look alike with other graphs across the manuscript as current authors’ contribution.

5) Reviewer: In the section 3, what types of methods were employed for the fabrication of nanomembranes? Please give a systematic summary for the generation methods of nanomembranes including their merits and demerits

Answer: The fabrication of nanomembranes is mentioned in the new table 1, and their advantages and limitations in the elaborated last section “Summary and conclusions”, accordingly. Our decision concerning these sections was dictated by our effort to reconcile all the overlapping referees’ suggestions in a balanced way.

6) Reviewer: Most recently, Electrospinning has been widely employed to fabricate nanofibrous filtration materials (10.1016/j.progpolymsci.2013.06.002, 10.1007/s10853-014- 8308-y and 10.1016/j.apmt.2022.101473). Is it suitable for the fabrication of nanomembranes for TPE application? Please add some more related discussion.

Answer: We mentioned this technology in “Summary and conclusions” and discussed it in the relevant context, accordingly. Our decision concerning this section was dictated by our effort to reconcile all the overlapping referees’ suggestions in a balanced way.

7) Reviewer: Some statements feel they are lacking references. Admittedly, some of these statement might be considered well known facts, the concepts mentioned might have been referenced previously or will be in the future, but it might still be pertinent add references next to these statements for new readers to the field, skim readers or people who don't necessarily want to go looking for the relevant reference.

Answer: We elaborated our reference list with 13 new citations which is influenced by this suggestion as well. Our decision concerning this section elaboration was dictated by our effort to reconcile all the overlapping referees’ suggestions in a balanced way.

8) Reviewer: The authors should make a critical review instead of plain text flow. This means that a comparative discussion should take place assisted with categorization of the attributes of the different systems in each section/Tables.
Answer: We did the suggested changes by giving a new table 1, and elaborating the last section “Summary and conclusions”, accordingly. Our decision concerning these sections was dictated by our effort to reconcile all the overlapping referees’ suggestions in a balanced way.

9) Reviewer: As well known that, nanomembrane-based strategies have been widely investigated for TPE application. Do they have any commercial products available now?

Answer: We elaborated the figure 4 by giving an additional picture of the commercial product in question.

10) Reviewer: The challenging and future direction in this area should be discussed in the conclusion section.

Answer: We included the relevant information in the last section “Summary and conclusions”, accordingly. 

11) Reviewer: The paper contains some typo and graphical errors. Please read carefully and correct Them.

Answer: As mentioned in our answer 4 above, we carefully checked all graphs and made the necessary corrections, accordingly

Reviewer 3 Report

The article Therapeutic Plasma Exchange in Certain Immune-Mediated 2 Neurological Disorders: Focus on a Novel Nanomembrane- 3 Based Technology is well written that provide a huge information about Nanomembrane-based TPE.

My major concern is about figure and table. In this article there are 16 Figs and 2 tables and all are adapted. There is not a single fig or table which was drawn by authors. So, it will be strongly recommended that authors apply their own idea and modify most of the figures.  

I am surprised to see that the authors have not included a section for methodology. Please add a heading for methodology in which they can provide a brief account on how they collect the data for their review.

Also discussed about the major limitation of Nanomembrane-based TPE therapy.

Write future direction and prospects of this study in a separate heading.

Conclusion section is very short.  Elaborate it.

Author Response

We would like to thank this reviewer for the useful remarks, which, no doubt, would improve the quality of this paper and would make it more readable and comprehensive.

We have taken into consideration every point raised in this review and have made the required changes as follows:

Reviewer: My major concern is about figure and table. In this article there are 16 Figs and 2 tables and all are adapted. There is not a single fig or table which was drawn by authors. So, it will be strongly recommended that authors apply their own idea and modify most of the figures.

Answer: We remodeled the graphs in order to look alike with other graphs across the manuscript as current authors’ contribution along with the table 1 and table 2 of all three tables after the revision (the table 3 should be presented as adapted one because of the economic estimates done by the cited authors).

Reviewer: I am surprised to see that the authors have not included a section for methodology. Please add a heading for methodology in which they can provide a brief account on how they collect the data for their review.

Answer: We introduced a new heading methodology, accordingly.

Reviewer: Also discussed about the major limitation of Nanomembrane-based TPE therapy. Write future direction and prospects of this study in a separate heading. Conclusion section is very short. Elaborate it.

Answer: We included the relevant information in the last section “Summary and conclusions”, accordingly. Our decision concerning this format (extended last section) was dictated by our effort to reconcile all the overlapping referees’ suggestions in a balanced way.

Reviewer 4 Report

The review is too short to be published. There are a little references in all section. The authors should perform an extended literature review and re-write the review with more sections, examples, references and figures. I do not recommend this review article to be published.

Author Response

We would like to thank this reviewer for the useful remarks, which, no doubt, would improve the quality of this paper and would make it more readable and comprehensive.

We have taken into consideration every point raised in this review and have made the required changes as follows:

Reviewer: The review is too short to be published. There are a little references in all section. The authors should perform an extended literature review and re-write the review with more sections, examples, references and figures.

Answer: We did all the necessary changes in the direction of more sections, more examples, more references and more tables and elaborated figures. Given the scope of our review – in certain immune-mediated neurological disorders and its focus on nanotechnology in rare auto-immune diseases, to our knowledge these are all published relevant data in the field until December 2022. We tried to fulfil the reviewer 4 suggestions as much as possible in the context of all the overlapping referees’ suggestions as well.

As a general answer to all four reviews – in line with the requirements of the Editor we have revised all texts highlighted in yellow using track changes, accordingly.

Hoping that the revised version would meet the requirements of Biomedicines for publication. 

Round 2

Reviewer 2 Report

The revised edition has improved a lot.

Reviewer 3 Report

Authors response are satisfactory. Accept in current form

Reviewer 4 Report

The authors have modified their manuscript significantly. However, more examples are required. A standard review consists of 200 or more references. This manuscript has only 102. Addition of more examples is necessary. Introduction part is too short. More background information having more references is required. Insertion of results in tabular form (at least 4 tables) is highly recommended. Authors should also check their English language.

Round 3

Reviewer 4 Report

The manuscript can now be accepted.